# Structure–Tissue Exposure/Selectivity Relationship (STR) on Carbamates of Cannabidiol

**DOI:** 10.3390/ijms252211888

**Published:** 2024-11-05

**Authors:** Sheng Wang, Jian-Guo Yang, Kuanrong Rong, Huan-Huan Li, Chengyao Wu, Wenjian Tang

**Affiliations:** Center for Scientific Research, School of Pharmacy, Anhui Medical University, Hefei 230032, China; wangsheng_cpu@163.com (S.W.); yjgyy777@163.com (J.-G.Y.); rongkuanrong@163.com (K.R.); lihh1028@163.com (H.-H.L.); chengyao.wu@hotmail.com (C.W.)

**Keywords:** structure–activity relationship (SAR), structure–tissue exposure/selectivity relationship (STR), drug development and optimization, cannabidiol, carbamate, butyrocholinesterase

## Abstract

The structure–tissue exposure/selectivity relationship (STR) aids in lead optimization to improve drug candidate selection and balance clinical dose, efficacy, and toxicity. In this work, butyrocholinesterase (BuChE)-targeted cannabidiol (CBD) carbamates were used to study the STR in correlation with observed efficacy/toxicity. CBD carbamates with similar structures and same molecular target showed similar/different pharmacokinetics. **L2** and **L4** had almost same plasma exposure, which was not correlated with their exposure in the brain, while tissue exposure/selectivity was correlated with efficacy/safety. Structural modifications of CBD carbamates not only changed drug plasma exposure, but also altered drug tissue exposure/selectivity. The secondary amine of carbamate can be metabolized into CBD, while the tertiary amine is more stable. Absorption, distribution, metabolism, excretion, and toxicity (ADMET) parameters can be used to predict STR. Therefore, STR can alter drug tissue exposure/selectivity in normal tissues, impacting efficacy/toxicity. The drug optimization process should balance the structure–activity relationship (SAR) and STR of drug candidates for improving clinical trials.

## 1. Introduction

At the final stage in drug discovery, the main goal of lead optimization is to attain a drug candidate that achieves a concentration–time profile in the body that is adequate for the desired efficacy and safety profile [1,2]. During lead optimization, structural modification not only changes pharmacokinetic properties in plasma, but also alters drug exposure/selectivity in disease-targeted tissues/normal tissue [3,4]. Despite great efforts to achieve lead optimization, the failure rate of drug development remains very high (~90%) [5,6]. However, overemphasis on drug exposure in plasma and overlooking drug distribution in disease-targeted tissue/normal tissues may mislead the selection of drug candidates for clinical trials [6,7,8,9,10].

The drug discovery process is long, complex, and expensive; thus, lead optimization can improve drug potency/specificity through the structure–activity relationship (SAR) and drug-like properties [4,11]. However, current lead optimization overly emphasizes potency/specificity using SAR but overlooks tissue exposure/selectivity in disease/normal tissues using the structure–tissue exposure/selectivity relationship (STR), which may mislead drug candidate selection and impact the balance of clinical dose/efficacy/toxicity. An STR study of selective estrogen receptor modulators suggested that STR alters drug tissue exposure/selectivity in disease-targeted tissues vs. normal tissues, impacting clinical efficacy/toxicity [12]. Lead optimization needs to balance the SAR and STR in selecting drug candidates for clinical trials to improve the success of drug development.

Carbamate can be hydrolyzed by AChE/BuChE in the blood to significantly affect oral bioavailability, but how does the substituent of carbamate influence pharmacokinetic and pharmacodynamic characteristics [11,12,13]? Rivastigmine, a pseudo-irreversible carbamate inhibitor, is a well-tolerated and effective treatment for Alzheimer’s disease (AD) [14,15]; however, the first-pass metabolism leads to its short plasma elimination half-life (1.5 h) and low bioavailability (36–40%), which is relative to cholinergic side effects like dyspepsia, anorexia, nausea, vomiting, and severe bradycardia [16,17]. The first-pass metabolism of physostigmine led to very low bioavailability (only 2%), and drug concentrations after oral administration were 200 times less in plasma and 350 times less in brains than after intramuscular administration [18,19]. Hence, the modification of carbamates for drug brain delivery can be conducted to improve the bioavailability of carbamate inhibitors [20,21,22,23].

Cannabidiol (CBD) is one of active constituents in *Cannabis sativa* plant and approved as a drug for seizure disorder [24,25]. Recently, CBD carbamates were designed to discover potent selective BuChE inhibitors, and an SAR study showed that the amine group of carbamates can regulate the selectivity and potential to ChE by molecular interactions with the active site of ChE [26,27]. The carbamate of CBD affected the selectivity and potency on BuChE inhibitory activity; for example, **L2** with methylethylamine showed similar ChE inhibitory activity to rivastigmine (IC_50_ values of AChE and BuChE of 14.95 and 0.077 μM, and 16.35 and 0.058 μM for **L2** and rivastigmine); **L1** and **L2** with aliphatic amine exhibited better BuChE inhibition than **L3** with cyclic amine; **L4** with *tert*-benzylamine showed the most potent BuChE-inhibitory activity (Figure 1). Therefore, it is significant to study the pharmacokinetics of CBD carbamates to guide structural modification and supply preclinical pharmacokinetic data.

Cholinergic side effects limit the clinical applications of pseudo-irreversible inhibitors [28,29]. In this study, a rapid and simultaneous ultra-performance liquid chromatography–high resolution mass spectrometer (UPLC-HRMS) method for determination of CBD (**L0**) and its carbamates (**L1**–**L4**) was established to investigate the comparative pharmacokinetics of **L0**–**L4**. The brain is the target tissue of BuChE inhibitors. STR can alter drug candidate’s tissue exposure/selectivity in normal organs, impacting efficacy/toxicity [30]. Comprehensive data on CBD carbamates can be used to carry out STR analysis. In addition, this study may contribute to the development of the longer-acting and selective candidates of pseudo-irreversible inhibitors with anti-AD. It is significant to know about how the SAR and STR of CBD carbamates affect pharmacokinetic and pharmacodynamic characteristics.

## 2. Results and Discussion

### 2.1. Drug Exposure in Tissue but Not in Plasma Was Associated with Efficacy/Toxicity

#### 2.1.1. No Correlation Between Drug Exposure in Plasma and in Disease-Targeted Tissues

In the drug candidate development process, drug exposure in plasma (concentration of the drug in plasma and the area under the curve, AUC) is often recognized as an evaluation of drug exposure in the disease-targeted tissue (such as brain). Drug candidates with high exposure in the plasma are often chosen for future drug-like studies [31]. To investigate how drug exposure in plasma correlates with drug exposure in disease-targeted tissues, combined with the inhibitory activities against *eq*BuChE and plasma exposure from AUC_plasma_, **L2** and **L4**, which shown better potent BuChE inhibitory activity and plasma exposure, were chosen for further invivo evaluation. We determined the drug concentration in plasma and target tissue (brain), and calculated the AUC after oral administration of **L2** and **L4** in rats (Figure 2A). Plasma AUC values of **L2** and **L4** were not correlated with AUCs in the brain tissue, as presented in Figure 2B,C. One interesting scenario was observed: similar drug plasma concentrations did not predict drug brain concentration. **L2** and **L4** had similar drug plasma concentration, whereas the brain concentration of **L2** was fivefold higher than that of **L4**.

For drug candidates targeting the central nerve system (CNS), drug exposure in brain tissue has been used for drug candidate selection in AD, Parkinson’s disease, and other CNS-related diseases [7,32,33]. Drug level in the brain or in cerebrospinal fluid (CSF) have been measured and considered as a selection criterion during CNS drug optimization [34,35]. Any drug candidate with no ability to reach the brain is terminated for further development [4,30]. Therefore, CNS drug optimization should adapt a criterion to ensure drug exposure in brain organ vs. normal organ [7].

#### 2.1.2. Structural Correlation of Drug Exposure with Drug Efficacy/Safety

To further study whether CBD drug exposure in the brain was better than plasma exposure, so as to correlate with drug efficacy/safety, we compared two CBD carbamates (**L2** and **L4**) with similar drug plasma exposure and evident efficacy/toxicity profiles. As shown in Table 1, **L2** with methylethylamine showed similar AChE inhibitory activity to rivastigmine; **L1** and **L2** with aliphatic amine exhibited better BuChE inhibition than **L3** with cyclic amine; **L4** with *tert*-benzylamine showed the most potent BuChE-inhibitory activity. **L2** and **L4** exposure in the plasma was >3-fold higher than CBD, **L1**, and **L3**. Interestingly, CBD, **L1**, and **L3** have close BuChE inhibition. Although **L2** and **L4** had similar drug exposure in the plasma, **L2** exposure in the brain was fivefold higher than **L4**, whereas BuChE inhibition of **L4** was more potent than that of **L2**.

ADMET analysis showed that compared to the rat acute oral toxicity (LD_50_) of CBD (319.5 mg/kg), LD_50_ values of **L2**–**L4** decreased to 70.5, 80.0, and 84.0 mg/kg, while that of **L1** significantly decreased to 22.1 mg/kg. Obviously, **L2** and **L4** had better safety profiles, but the secondary amine of carbamate (**L1**) increased oral toxicity, indicating that the amine group of carbamate affected its exposure in the plasma, which may be associated with drug efficacy/safety.

#### 2.1.3. Relationship of Drug Exposure in Tissue and Plasma

Drug exposure in tissue is calculated by drug exposure in the plasma and tissue/plasma distribution coefficient (*K*_p_), as shown in Equation (1):Drug exposure in the tissue = drug exposure in the plasma × *K*_p_(1)
where drug exposure in the plasma and the tissue can be calculated by the drug concentration vs. time curve, and *K*_p_ values can be calculated by *C*_tissue_/*C*_plasma_ or AUC_tissue_/AUC_plasma_ [36]. Based on the inhibitory activities against *eq*BuChE and plasma exposure from AUC_plasma_, **L2** and **L4** were chosen for further invivo evaluation. As shown in Table 2, *K*_p_ values (AUC_tissue_/AUC_plasma_ ratio) of **L2** and **L4** in tissues showed significant difference although they had similar plasma AUCs. Hence, it was unreasonable to use drug exposure in the plasma to substitute for drug exposure in the disease-targeted tissue due to neglect of the differences of *K*_p_ values in various tissues for drug candidates. It is worth mentioning that *K*_p_ can be evaluated by the ratio of free fraction of drug candidate in plasma vs. in tissues in steady state. Nevertheless, this assessment is based on “free drug hypothesis” theory, which presumes that free drug concentration in tissue is close to that in plasma [37].

In the BuChE-targeted drug optimization process, an ideal drug candidate not only has high specificity and potency against BuChE targets with low *K*_i_ (or low IC_50_), but also has high tissue exposure/selectivity in the brain to exert efficacy, and low tissue exposure/selectivity in other tissues to reduce toxicity [4,30].

In this study, drug candidates **L2** and **L4** with high plasma exposure were selected for further exposure/selectivity studies; however, if low tissue exposure with low tissue toxicity has a high maximum tolerated dose (MTD), a high dose is often required to achieve efficacy in clinical trials. Although **L2** and **L4** had similar exposure in plasma, both exposure and selectivity in the tissues had very different associations with their clinical efficacy/toxicity profiles (Figure 1 and Figure 2 and Table 3). Therefore, due to the lack of drug exposure/selectivity in the brain, the high CNS drug exposure/selectivity in healthy vital organs may lead to unmanageable toxicity in clinical studies [4].

### 2.2. Drug Tissue Selectivity May Impact the Balance of Efficacy/Toxicity, Which Is Often Overlooked in the Drug Optimization Process

Dose escalation tests are always evaluated during clinical phase I/II trials, and MTD is constantly associated with drug exposure in certain toxicity-related organs [38]. Dose escalation results in sufficient drug exposure in disease-targeted tissues, whereas the probability of adverse effects for drug exposure in blood cells or vital organs also increases. To evaluate the balance between efficacy and toxicity for drug candidates, drug tissue selectivity is a crucial parameter and defined as follows in Equation (2), where the sum of *C*_tissue_ or AUC_tissue_ is total drug concentration or AUC in different tissues.
Drug tissue selectivity = *C*_tissue_/Σ*C*_tissue_ or AUC_tissue_/ΣAUC_tissue_(2)

A perfect CNS drug candidate is supposed to exhibit high tissue selectivity and exposure in the brain for better therapeutic efficacy, and low tissue selectivity and exposure in other vital organs to reduce toxicity in the meantime. On the contrary, if a CNS drug candidate has low selectivity and low exposure in the brain but high selectivity/exposure in other vital organs, it may not be able to reach its therapeutic concentration in the CNS. Moreover, if a drug candidate shows low selectivity and exposure in both the brain and other vital organs, this indicates that the drug candidate maybe safe even with high drug exposure in the plasma during preclinical studies, but it may present insufficient therapeutic efficacy in further clinical studies. Drug candidate innovators should pay ample attention to this phenomenon, and this dilemma should be avoided during drug optimization.

As shown in Figure 3A,B, the tissue exposure and selectivity were compared between **L2** and **L4**. Despite the similar plasma exposure of **L2** and **L4**, **L2** had 5.1-fold higher exposure than **L4** in the brain, and **L2** had 1.8- to 2.8-fold lower exposure than **L4** in the liver and kidney, respectively. All in all, **L2** tissue selectivity in the brain was 21-fold higher than that of **L4**, whereas **L2** tissue selectivity in the liver and kidney was 1.8- and 1.2-fold lower than that of **L4**, respectively (Figure 3B). Due to differing tissue selectivity, a much higher dose of **L4** may achieve similar drug exposure in the brain compared to **L2**, but this would lead to higher **L4** exposure in the liver and kidney to increase the drug toxicity in metabolism and elimination organs in the body.

In present study, three important PK parameters for studying STR, drug exposure, partition coefficient *K*_p_, and drug selectivity in tissues, were closely associated with drug efficacy/toxicity. The content of drug candidate accumulated in certain tissue (Table 2) was evaluated by drug exposure (AUC_tissue_) and drug candidate partition coefficient (*K*_p_) in the tissues, while drug tissue selectivity may impact the drug candidate’s therapeutic window between efficacy and toxicity (Figure 3).

Even in normal organs, the “free drug hypothesis” may not be accurate due to the transportation of both protein-bound drugs and free unbound drugs in normal tissues and disease-targeted tissues [7,32,33]. The overemphasis of free drug content in the plasma and neglect of drug tissue exposure may misguide the optimization process and selection of drug candidate. It is essential to choose total drug tissue exposure or *K*_p_ (total drug in tissue/plasma ratio) for drug candidate selection in lead optimization [4,9,10,39,40]. Hence, the adjustment of dose regimen for **L2** and **L4** should be carefully considered in terms of drug exposure in the brain/liver rather than in plasma exposure for clinical therapeutic efficacy, particularly regarding their unique activity as BuChE inhibitors.

### 2.3. Structural Modification Altered Drug Exposure and Selectivity in Various Tissues Despite Similar Drug Exposure in the Plasma

Structural modification of leads is a common technique in pharmacochemistry for lead optimization, owing to the fact that modification may impact the binding affinity of drug candidates to the molecular targets and pharmacokinetics of the drug candidate in the body. However, it is unclear that these structural modifications may alter drug exposure/selectivity in the tissues, which may affect their clinical efficacy/toxicity. Thus, we compared the drug exposure or tissue selectivity in various tissues of five CBD carbamates with similar chemical structures: **L0**, **L1**, **L2**, **L3**, and **L4**.

After oral administration of CBD and its carbamates, we firstly compared the difference in plasma concentration. The data revealed that drug plasma exposure and selectivity were significantly altered after structural modification. **L1** and **L3** had almost the same drug exposure as CBD in the plasma, which was much lower compared to **L2** and **L4**. **L2** and **L4** had similar and the largest drug exposure in the plasma, but there was distinct tissue exposure in most organs, such as the brain, heart, kidney, liver, lung, and spleen. **L2** and **L4** had 0.08-fold to 16-fold exposure differences in tissues.

Moreover, the difference for tissue selectivity of **L2** and **L4** are presented in Figure 4. **L2** had higher selectivity in the brain, kidney, and liver, while **L4** had higher selectivity in the heart, kidney, and lung. These data prove that structural modifications can obviously change drug exposure and selectivity in different tissues, even though they exhibit similar or same drug exposure in plasma. However, these phenomena are always overlooked during drug selection in the drug optimization process.

### 2.4. ADMET and Physicochemical Property Prediction of CBD Carbamates

Although STR may provide guidance and orientation for lead optimization, the limited knowledge on STR relationship obstructs our understanding. Drug exposure in the plasma and tissues may be influenced by physicochemical properties, such as chemical solubility, lipophilicity (log*P*), ionization (p*K*_a_), polarity (such as polar surface area, PSA), plasma protein/tissue binding, and molecular weight (MW) [41,42]. However, CBD and **L1**–**L4** in the present study had very similar physiochemical properties (Table 4 and Appendix A). For example, **L2** and **L4** had close TPSA, log*P*, and MW, but both exhibited as large as fivefold differences in drug exposure in various tissues. Therefore, physicochemical properties cannot accurately explain the differences in drug exposure and selectivity in tissues. To better illuminate the molecular structure descriptors of these drug candidates, which may be related to drug exposure and tissue selectivity in different tissues, we used Mat-ADMET to collect more than 60 molecular structure descriptors to predict the ADMET of the substances (Appendix A) to rationalize the results of the STR.

The results in Table 4 and Appendix A show that CBD and **L1**–**L4** complied with the rule-of-five (RO5) of orally administered drug candidates (four simple physicochemical parameter ranges: MW ≤ 500, log*P* ≤ 5, H-bond donors ≤ 5, and H-bond acceptors ≤ 10), 90% of which have accomplished clinical phase II trials; they might had blood–brain barrier (BBB) permeability and proper plasma protein binding rate, in which **L2** had the best BBB permeability. CBD and **L1**–**L4**, the number of N and O atoms ≤ 5 in a molecule or log*P* − (N + O) > 0 had high potential to reach the brain as CNS drug candidates. CBD had long a half-life (*t*_1/2_). Interestingly, the introduction of a fat chain (**L1** and **L2**) led to *t*_1/2_ decrease; the introduction of pyrrolidine (**L3**) and halogen-containing benzyl (**L4**) led to *t*_1/2_ increase [43]. Further, TPSA values of CBD and **L1**–**L4** (<60) tended to be identified as CNS-active drug candidates. **L2** had the lowest P-glycoprotein inhibition related to CNS transporter affinity. Carbamate decreased the rat acute oral toxicity (LD_50_) of CBD; however, **L2**–**L4** had higher LD_50_ values than **L1**, which showed that the tertiary amine of carbamate had lower oral toxicity. These predictions indicated that CBD carbamates, especially **L2**, had a benign safety profile, and may be developed as a CNS-active drug candidate.

Therefore, it is worth considering the importance of STR in the chemical efficacy/toxicity during the drug optimization process. However, more data are required to further understand how the structural properties of carbamates relate to brain tissue. As a proof of concept, univariate feature analysis was used to study whether *K*_p_ is correlated with different molecular properties. The correlation of structure descriptors with the brain and liver were observed in CBD and four carbamates; these preliminary studies are required rational STR studies for drug candidate design in future [4,44,45]. It is crucial to keep a balanced structure–tissue exposure/selectivity–activity relationship (STAR) for drug candidate selection and drug optimization processes [4,30], which contributes to improving the clinical drug development.

### 2.5. Pharmacokinetics Study of CBD and Its Carbamates **L1**–**L4**

PD and PK studies are an integral part of lead optimization. Lead optimization is the complex, non-linear process of refining the chemical structure of a confirmed drug candidate to improve its drug-like properties. CBD carbamates had a more predictable response, greater bioavailability, and more potent BuChE inhibition than CBD. However, the amine substituent of carbamate affected their physicochemical and biological properties, as well as PK characteristics.

A rapid and sensitive UPLC-HRMS/MS method was established for the simultaneous determination of CBD and its carbamates **L1**–**L4** in rat plasma samples for PK study. As shown in Table 3 and Figure 5, relevant PK parameters were obtained using WinNonLin software. The *t*_1/2_ and *T*_max_ of CBD were 3.70 h and 1.20 h, respectively, which are consistent with the reported values of 1.4–10.9 h and 0–4 h [46,47]. The *T*_max_ of **L2** increased slightly to 1.80 h, while those of **L1**, **L3**, and **L4** decreased to 0.22 h, 0.95 h, and 0.80 h, respectively, indicating that CBD carbamates can be rapidly metabolized in vivo. In the present work, carbamates reduced the long half-life (*t*_1/2_) of CBD to proper values (Table 4), but the *C*_max_ after transformation greatly increased, that is, after the introduction of amine groups into CBD, the blood concentration increased. For example, **L4** increased significantly to 210.89 μg/L; **L1** and **L2** increased to 119.98 μg/L and 127.03 μg/L, respectively; and **L3** increased slightly to 60.01 μg/L. Further, **L2** and **L4** exposure in the plasma were similar: their AUC_0-T_ values were 561.35 μg·h/L and 521.56 μg·h/L, respectively. Moreover, **L2** and **L4** had similar metabolisms, their *V*_d_ and CL values were 99.38 L/kg and 26.07 L/h/kg, and 91.52 L/kg and 28.99 L/h/kg, respectively. However, **L2** and **L4** exposures in the brain were significantly different.

Further analysis of blood samples showed that CBD was observed in the plasma of **L1**, indicating that only **L1** can be metabolized to produce CBD (Figure 5). Therefore, the tertiary amine of carbamate may be more stable than the secondary amine for CBD carbamate. Further, the *C*_max_ and AUC_0-T_ of **L2** and **L4** in the brain were 56 ng/g and 19 ng/g, and 213.8 ng·h/g and 43.2 ng·h/g, respectively, indicating that **L2** had much higher exposure than **L4** in the brain (Figure 2C and Table 2). The data of Mat-ADMET also showed that compared with **L2**, benzyl in **L4** led to a higher plasma protein binding rate and lower brain exposure (Table 3). Therefore, the pharmacokinetics of carbamates was significantly affected by the chemical structures, which may guide structural modification [48,49,50].

## 3. Materials and Methods

### 3.1. Chemicals and Reagents

CBD was purchased from Push Bio-technology (Chengdu, China). The CBD carbamates (**L1**–**L4**), internal standard [**IS**, **L5**, CBD 4-bromobenzyl(methyl)carbamate] were synthesized, and the CBD **L0**–**L5** were provided by School of Pharmacy, Anhui Medical University (Hefei, China). Methanol (MeOH), acetonitrile (ACN), ammonium formate, and formic acid (FA) of LC–MS grade were purchased from Thermo Fisher (San Jose, CA, USA). HPLC-grade ammonium acetate, methyl tertiary butyl ether (MTBE), ethyl acetate (EAC), and trichloromethane (CHCl_3_) were obtained from Sigma-Aldrich (St. Louis, MO, USA). Other reagents were chemically pure, and the deionized water was purified using a Milli-Q-Plus system (Bedford, MA, USA).

### 3.2. Animal Experiments

Animal experiments were conducted in accordance with the National Institutes of Health Laboratory Animal Care and Use Guidelines. Male SPF SD rats from the Animal Center of Anhui Medical University (Hefei, China) were used for the experiment. Compounds **L0**–**L4** were dissolved in ethanol, and the single intragastric administration to rats at a dose of 15 mg/kg. At each time point (0.5, 1, 2, 4, and 8 h) after administration, samples of brain, heart, kidney, liver, lung, and spleen were collected from each rat for tissue distribution study. While for pharmacokinetic study, the plasma samples were gathered from the suborbital vein at 5, 15, 30, and 45 min and 1, 2, 4, 8, and 12 h after oral administration. The maintenance and treatment procedures for experimental animals were approved by the Animal Care and Use Committee of Anhui Medical University (approval number: LLSC20190274). Room temperature was 22 ± 2 °C, using a light/dark (12: 12 h) cycle, the animals were free to consume food and water one day before the experiment.

### 3.3. Samples Preparation Procedures

An aliquot of 60 μL plasma sample was spiked with 60 µL aqueous ascorbic acid (0.1%) and 20 μL of IS working solution (300 ng/mL). After vortexing for 30 s, 400 µL MTBE was added to plasma sample for liquid–liquid extraction. The mixture was vigorously mixed for 10 min and centrifuged at 12,000 rpm at 4 °C for 5 min; approximately 400 µL of supernatant was transferred to a clean 1.5 mL centrifuge tube and dried under a gentle stream of nitrogen gas. Subsequently, the residue was reconstituted with 100 µL of MeOH/water (7: 3, *v*/*v*), vortex-mixed for 2 min, and followed by centrifugation at 12,000 rpm for 20 min. Finally, 20 μL of the supernatant was injected into the UPLC-HRMS system for measurement. The tissue sample (100 mg) was homogenized in normal saline solution (1 mL), 600 μL homogenate sample was spiked with 60 µL aqueous ascorbic acid (0.1%) and 50 μL of the IS working solution (1000 ng/mL), and 1.2 mL MTBE was added for extraction. Then, 1 mL of supernatant was dried and proceeded as plasma sample pretreatment (Appendix A).

### 3.4. UPLC–HRMS Analysis of Drug Candidate Concentration

Quantitative analysis using Dionex Ultimate 3000 chromatography hyphenated to Q-Exactive plus Orbitrap mass spectrometer system (Thermo Fisher Scientific, Waltham, MA, USA). The analytes were detected in Full MS/dd-MS^2^ mode to choose the parent ion and optimize the retention time; then, we used the parallel reaction monitoring (PRM) mode through positive ionization condition for determination. Xcalibur 4.1 software (Thermo Fisher Scientific) was used for data acquisition and processing.

Chromatographic separation was achieved on a Shimadzu-pack GIST column (50 mm × 2.1 mm, 2 μm, Shimadzu Company, Kyoto, Japan). Binary mobile solvents consisted of water (containing 0.1% FA, *v*/*v*) (A) and acetonitrile (B), the gradient program was as follows: 0 min 60% A → 1 min 60% A → 1.5 min 20% A → 7 min 20% A → 7.1 min 60% A → 8.5 min 60% A. The flow rate was set at 0.30 mL/min and the column temperature was 40 °C. The typical source parameters were optimized as follows: spray voltage: 4500 V; capillary temperature: 320 °C; auxiliary gas heater temperature: 200 °C; pressure of sheath gas: 320 kPa; auxiliary gas: 50 kPa. The retention time and the optimal mass parameters are shown in Appendix A. The method was validated according to U.S. Food and Drug Administration (FDA) guidance for selectivity (Appendix A), linearity (Appendix A), matrix effect, recovery (Appendix A), precision and accuracy (Appendix A), and stability (Appendix A).

### 3.5. ADMET and Physicochemical Properties Prediction of CBD Carbamates

Mat-ADMET is a computational tool used to evaluate the absorption, distribution, metabolism, excretion, and toxicity (ADMET) profiles, along with physicochemical properties of compounds. Mat-ADMET enables us to forecast various properties such as lipophilicity, water solubility, physicochemical, pharmacokinetics, drug-likeness, and other pertinent descriptors. Additionally, it provides valuable insight into predicting the similarity of leads towards cytotoxicity, carcinogenicity, hepatotoxicity, mean lethal dose (LD50) in rodents, and other parameters. The Mat-ADMET website is accessible freely using following the link: Mat-ADMET (matwings.com (accessed on 12 December 2022)).

### 3.6. Pharmacokinetics

Phoenix/WinNonlin software (version 6.4; Pharsight, Mountain View, CA, USA) was used to calculate the AUC and other pharmacokinetic parameters in plasma or tissue for each drug in a non-compartmental model.

### 3.7. Statistical Analysis

GraphPad Prism (version 8.0.1) software was used for statistical analysis. Statistical significance between two groups was calculated by Student’s *t*-test; the *p* value < 0.05 was considered significant.

## 4. Conclusions

During drug optimization process, the STR can improve drug selection and influence the balance of clinical efficacy/toxicity. In the present study, we investigated the STR in correlation with drug efficacy/toxicity using BuChE-targeted CBD and its carbamates **L1**–**L4** with similar structures, the same molecular target, and similar/different PK profiles in plasma and tissues. The results showed that CNS drug exposure in the plasma was not correlated with those in the target tissue (brain), while tissue exposure/selectivity was correlated with drug efficacy/safety. **L2** and **L4** had similar drug exposures in the plasma, and **L2** showed higher exposure and selectivity in the brain than **L4**, which was likely due to plasma protein binding and P-glycoprotein inhibition. In addition, structural modifications not only changed drug exposure in the plasma, but also altered drug exposure and selectivity in various tissues. The tertiary amine of carbamate (**L2**–**L4**) was more stable than the secondary amine (**L1**) in the plasma. Further, the ADMET study showed that Mat-ADMET can predict STR. The data indicated that STR can alter drug tissue exposure/selectivity in organic tissues, impacting clinical efficacy/toxicity. Owing to the limitation of the AD animal model, drug exposure in disease tissues may induce different STRs. The drug optimization process should balance SAR and STR in improving clinical trials.

## Figures and Tables

**Figure 1 ijms-25-11888-f001:**
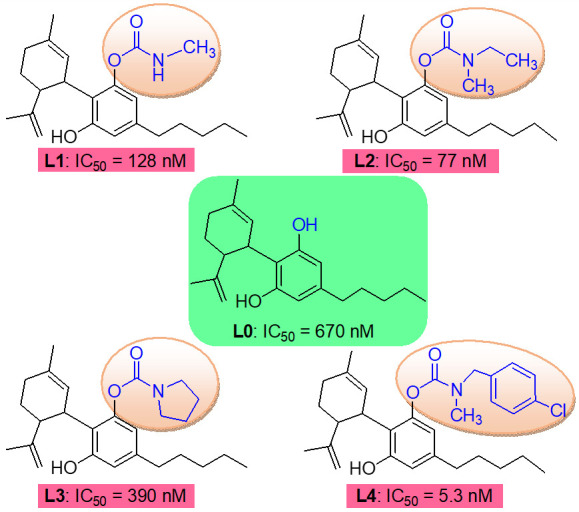
Chemical structures of CBD and its carbamates **L1**–**L4** used in this study. Compound inhibits the activity of the enzymes acetylcholinesterase (AChE) or BuChE, which catalyzes the degradation/hydrolysis of the neurotransmitter acetylcholine. IC_50_ values were defined as the concentration of compound where percent inhibition rate on the BuChE inhibition is equal to 50 and was the mean from three independent experiments in this study.

**Figure 2 ijms-25-11888-f002:**
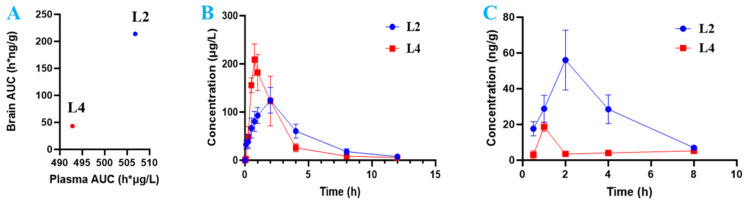
(**A**) The drug concentration and AUC in plasma and target tissue (brain) after oral administration of **L2** and **L4** in rats: plasma AUC vs. tissue AUC of brain. (**B**) Concentration–time curve of **L2** vs. **L4** in the plasma. (**C**) Concentration–time curve of **L2** vs. **L4** in the brain.

**Figure 3 ijms-25-11888-f003:**
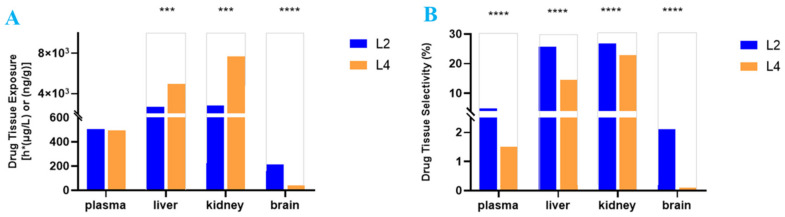
Drug tissue selectivity is a critical parameter that tips the balance of efficacy/toxicity. Comparison of drug exposure (**A**) and selectivity (**B**) in brain, liver, and kidney tissues between **L2** and **L4** after oral administration (15 mg/kg), *** *p* < 0.05, **** *p* < 0.01.

**Figure 4 ijms-25-11888-f004:**
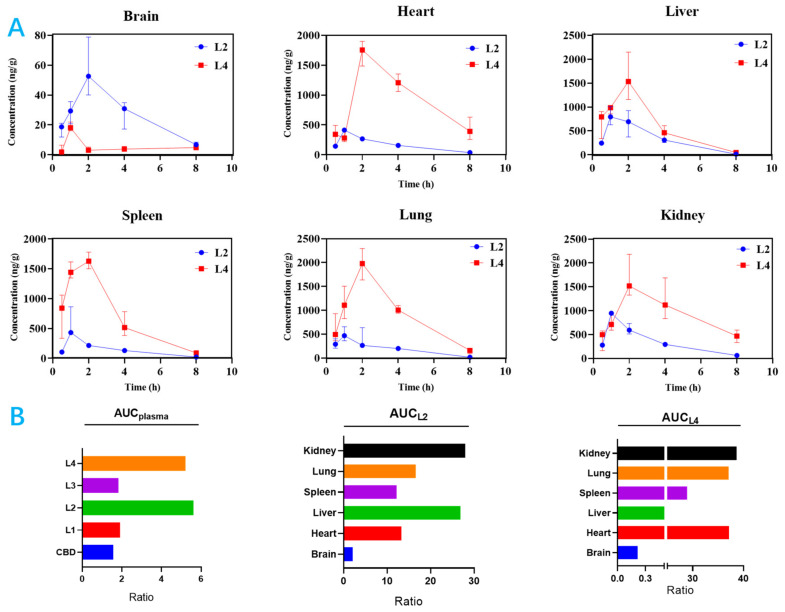
Structural modification alters drug exposure and selectivity in tissues despite similar exposure in plasma. (**A**) Concentration–time curve of **L2** vs. **L4** in tissues. (**B**) AUC ratio of **L2** vs. **L4**.

**Figure 5 ijms-25-11888-f005:**
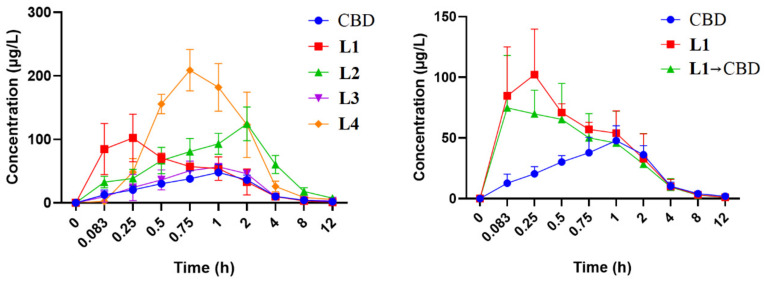
Average blood concentration–time curve of each analyte (n = 5).

**Table 1 ijms-25-11888-t001:** Plasma AUC (ng·h/mL) after oral administration on rats and inhibitory activities against *ee*AChE and *eq*BuChE of CBD and **L1**–**L4** (Ref. [27]) *^a^*.

Compound	AUC_plasma_ (ng·h/mL)	IC_50_, µM (or Inhibition Rate % at 20 µM)	
		AChE *^b^*	BuChE *^c^*
CBD	157.5	17.07 ± 2.43	0.67 ± 0.06
**L1**	191.2	18.88 ± 1.11	0.128 ± 0.022
**L2**	561.4	14.95 ± 1.02	0.077 ± 0.005
**L3**	182.1	*na ^d^*	0.39 ± 0.04
**L4**	521.6	21.4 ± 2.8%	0.0053 ± 0.0012
rivastigmine		16.35 ± 1.54	0.058 ± 0.013

*^a^* Each IC_50_ value is the mean ± SEM from at least three independent experiments; thereinto, ChE inhibitors inhibit the activity of the enzymes AChE and BuChE, which catalyze the degradation/hydrolysis of the neurotransmitter acetylcholine (ACh); *^b^ ee*AChE from electric eel; *^c^ eq*BuChE from horse serum; *^d^ na*: no activity.

**Table 2 ijms-25-11888-t002:** Plasma and tissue AUC (ng·h/mL or ng·h/g) after oral administration of **L2** and **L4** in rats and *K*_p_ values were calculated by AUC_tissue_/AUC_plasma_.

Parameter	L2	L4
AUC_plasma_	506.8	492.8
AUC_brain_	213.8	43.2
AUC_heart_	1331.9	7431.3
AUC_liver_	2684.5	4951.2
AUC_spleen_	1220.9	5773.9
AUC_lung_	1655.0	7421.8
AUC_kidney_	2790.2	7723.4
*K* _p brain_	0.42	0.088
*K* _p heart_	2.63	15.08
*K* _p liver_	5.30	10.05
*K* _p spleen_	2.41	11.72
*K* _p lung_	3.27	15.06
*K* _p kidney_	5.51	15.67

**Table 3 ijms-25-11888-t003:** Physicochemical properties of CBD carbamates with similar structures. The data come from Mat-ADMET.

Name	CBD	L1	L2	L3	L4
MW	314.47	371.53	399.58	411.59	496.09
TPSA	40.46	58.56	49.77	49.77	49.77
Log*P*	6.455	5.887	6.710	6.913	8.053
Log*D*	4.792	4.111	4.181	4.076	4.817
Log*S*	−4.654	−4.300	−3.840	−4.061	−5.751
Solubility	0.010	0.014	0.021	0.017	0.003
BBB penetration	0.915	0.804	0.957	0.921	0.902
*t*_1/2_ (hours)	8.091	6.094	6.015	8.595	8.273
Pgp inhibitor	0.128	0.878	0.102	0.994	1.000
Rat acute oral toxicity (LD_50_)	319.474	22.061	70.497	80.045	84.020

**Table 4 ijms-25-11888-t004:** Pharmacokinetic parameters of CBD and **L1**–**L4** in plasma after intragastric administration.

Pharmacokinetic Parameters	CBD	L1	L2	L3	L4	L1→CBD
*t*_1/2_ (h)	3.70 ± 1.26	2.70 ± 1.27	2.64 ± 0.45	3.09 ± 1.52	2.17 ± 0.19	3.43 ± 1.64
*T*_max_ (h)	1.20 ± 0.45	0.22 ± 0.07	1.80 ± 0.45	0.95 ± 0.11	0.80 ± 0.11	0.27 ± 0.15
*C*_max_ (μg/L)	48.89 ± 11.02	119.98 ± 15.90	127.03 ± 23.65	60.01 ± 8.63	210.89 ± 30.74	94.59 ± 27.06
AUC_0-T_ (μg·h/L)	157.50 ± 10.61	191.20 ± 58.69	561.35 ± 104.32	182.10 ± 13.94	521.56 ± 133.68	165.91 ± 87.56
AUC_0-∞_ (μg·h/L)	168.17 ± 11.50	196.80 ± 57.97	590.36 ± 110.07	200.40 ± 32.51	539.00 ± 131.18	172.51 ± 84.52
MRT_0-T_ (h)	2.84 ± 0.23	2.05 ± 0.47	3.44 ± 0.22	2.72 ± 0.60	2.38 ± 0.21	2.21 ± 0.42
*V*_d_ (L/kg)	474.16 ± 155.23	319.94 ± 171.55	99.38 ± 23.95	320.94 ± 106.75	91.52 ± 22.81	577.46 ± 442.26
CL (L/h/kg)	89.55 ± 6.43	81.20 ± 21.71	26.07 ± 4.49	76.37 ± 11.68	28.99 ± 6.02	103.13 ± 42.44

## Data Availability

The authors do not have permission to share data.

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
