# Peer review of "Structure–Tissue Exposure/Selectivity Relationship (STR) on Carbamates of Cannabidiol"

_ijms, 2024, doi:10.3390/ijms252211888_

Round 1

Reviewer 1 Report

Comments and Suggestions for Authors

ijms-3297260

            The paper entitled “Structure‒tissue exposure/selectivity relationship (STR) on carbamates of cannabidiol” deals with the butyrocholinesterase (BuChE)-targeted cannabidiol (CBD) carbamates L0-L4 (5 products) and evaluated their structure‒tissue exposure/selectivity relationship (STR) with observed efficacy/toxicity. Structurally L2 and L4 being different with methylethylamine and 4-chlorobenzyl substituents had almost the same plasma exposure, but L2 had higher exposure in the brain.

Structurally these L2 and L4 are tertiary amine which were more stable compared to secondary carbamate which can be metabolized to CBD. Therefore, the Absorption, distribution, metabolism, excretion, and toxicity (ADMET) parameters were evaluated to predict STR. The STR can alter the drug’s tissue exposure/selectivity in normal tissues impacting efficacy/toxicity. Therefore, such drug optimization process should balance the structure-activity relationship (SAR) and STR of drug candidates for improving their outcomes in clinical trials.

Comments-

Figure1- the legend is not clear on which of the IC50 values are being displayed in the figure. From table 1, it was understood that these are the BuChE inhibition values.

In the results and discussion, it should be clearly mentioned that based on the inhibitory activities against eqBuChE of CBD and L1‒L4, the most active derivatives were chosen for further in-vivo evaluation. This explanation is mentioned in section 3.1.2.

Since this study design was chosen on the basis of drug level in brain it forms an important selection criterion during the CNS drug optimization. It is also important to understand the drug exposure in brain organ vs. normal organ, during CNS drug optimization. This study has helped to understand the adjustment of the dose regimen for L2 and L4 to consider the drug exposure in the brain/liver rather than in plasma exposure for clinical therapeutic efficacy in particular to their unique activity as BuChE inhibitors.

Correct line 184-185-

As shown in Table 1, L2 with methylethylamine showed similar AChE inhibitory activity to rivastigmine;

Acute toxicity study provided the safety profile for these derivatives L2- L4 and the structural difference were noted for the higher toxicity of L1.

Correct line 257-258- L2 had 5.1-fold higher 257 exposure than L4 in brain,

I feel the section presented as 3.3 should be written together with 3.1.1, the missing link of structure and drug concentration exposure in plasma will make sense.

Author Response

The paper entitled “Structure‒tissue exposure/selectivity relationship (STR) on carbamates of cannabidiol” deals with the butyrocholinesterase (BuChE)-targeted cannabidiol (CBD) carbamates L0-L4 (5 products) and evaluated their structure‒tissue exposure/selectivity relationship (STR) with observed efficacy/toxicity. Structurally L2 and L4 being different with methylethylamine and 4-chlorobenzyl substituents had almost the same plasma exposure, but L2 had higher exposure in the brain.

Structurally these L2 and L4 are tertiary amine which were more stable compared to secondary carbamate which can be metabolized to CBD. Therefore, the Absorption, distribution, metabolism, excretion, and toxicity (ADMET) parameters were evaluated to predict STR. The STR can alter the drug’s tissue exposure/selectivity in normal tissues impacting efficacy/toxicity. Therefore, such drug optimization process should balance the structure-activity relationship (SAR) and STR of drug candidates for improving their outcomes in clinical trials.

Comments-

Figure1- the legend is not clear on which of the IC50 values are being displayed in the figure. From table 1, it was understood that these are the BuChE inhibition values.

Response: We feel great thanks for your professional review work on our article. We added explanation in the footnote for Figure1 in line 82.

In the results and discussion, it should be clearly mentioned that based on the inhibitory activities against eqBuChE of CBD and L1‒L4, the most active derivatives were chosen for further in-vivo evaluation. This explanation is mentioned in section 3.1.2.

Response: Thank you very much for your helpful suggestion, we have added the explanation in line 214-215. “Based on the inhibitory activities against eqBuChE and plasma exposure from AUCplasma, L2 and L4 were chosen for further in-vivo evaluation.”

Since this study design was chosen on the basis of drug level in brain it forms an important selection criterion during the CNS drug optimization. It is also important to understand the drug exposure in brain organ vs. normal organ, during CNS drug optimization. This study has helped to understand the adjustment of the dose regimen for L2 and L4 to consider the drug exposure in the brain/liver rather than in plasma exposure for clinical therapeutic efficacy in particular to their unique activity as BuChE inhibitors.

Response: Thanks for your advice, this is a good idea. The drug exposure in the brain/liver may reflect the clinical therapeutic efficacy in particular. The calculation results showed that L2 has lower the value of brain/liver than L4. This will help us to our study in the future.

Correct line 184-185

As shown in Table 1, L2 with methylethylamine showed similar AChE inhibitory activity to rivastigmine;

Response: Thank you very much for your helpful suggestion, we have modified this point to “As shown in Table 1, L2 with methylethylamine showed similar AChE inhibitory activity to rivastigmine” in line 189.

Acute toxicity study provided the safety profile for these derivatives L2- L4 and the structural difference were noted for the higher toxicity of L1.

Correct line 257-258- L2 had 5.1-fold higher 257 exposure than L4 in brain,

Response: We sincerely thank the reviewer for careful reading. As suggested by the reviewer, we have corrected this mistake in line 264.

I feel the section presented as 3.3 should be written together with 3.1.1, the missing link of structure and drug concentration exposure in plasma will make sense.

Response: Thanks for your advice. We fell that there is a progressive relationship from 3.1 to 3.3, which is more reasonable.

Reviewer 2 Report

Comments and Suggestions for Authors

The submitted manuscript presents the original works on the pharmacokinetic studies of the derivatives (carbamates, total four of them) of cannabidiol, an important API. The study involves both in vivo (animals) and in silico (ADMET prediction) analysis. This work is of high quality, properly presented, with sufficient amount of newly presented results. The methods are properly described too. Therefore, I honestly recommend to publish this work after minor revisions.

While the authors focus on the pharmacokinetics, distribution and metabolism, the should have consider the possible pharmacodynamic aspects as well. I mean, the binding modes of the studied L1-L4 derivatives should be checked using the molecular modelling methods, i.e. molecular docking combined with the molecular dynamics simulations.

Line 35, SAR, Line 37, STR, acronyms should be explained there. I’m aware that the authors have done it in the abstract, but this should be done the first time they appears in the main text as well

Line 77, at this point the authors should state the aim of the study in a concise way

Line 81, it should be “values were”, also, it should be stated whether those values have been obtained as a part of the current study or they are simply cited

Line 86, did you really synthesize the CBD (L0)? Wasn’t it more convenient to purchase this reagent?

Lines 157-180, it should be clearly stated why L1 and L3 haven’t been compared

Figure 2C, why there are no error bars for L4? The same applies to Figure 4A.

I really appreciate the use of the other (blue) color for the references. However, I’m afraid this will be changed to black at the production stage…

Author Response

The submitted manuscript presents the original works on the pharmacokinetic studies of the derivatives (carbamates, total four of them) of cannabidiol, an important API. The study involves both in vivo (animals) and in silico (ADMET prediction) analysis. This work is of high quality, properly presented, with sufficient amount of newly presented results. The methods are properly described too. Therefore, I honestly recommend to publish this work after minor revisions.

While the authors focus on the pharmacokinetics, distribution and metabolism, the should have consider the possible pharmacodynamic aspects as well. I mean, the binding modes of the studied L1-L4 derivatives should be checked using the molecular modelling methods, i.e. molecular docking combined with the molecular dynamics simulations.

Response: Thanks for your advice. Mat-ADMET was used to collect more than 60 molecular structure descriptors to predict ADMET. We believed that ADMET analysis is more suitable for the study on pharmacodynamic aspects, while the molecular modelling is better for new lead compounds.

Line 35, SAR, Line 37, STR, acronyms should be explained there. I’m aware that the authors have done it in the abstract, but this should be done the first time they appears in the main text as well.

Response: Thanks for your suggestion. We have added the full name before Line 35, SAR, Line 37, STR.

Line 77, at this point the authors should state the aim of the study in a concise way

Response: Thanks for your valuable advice. We have changed the expression to “It is significant to know about how the SAR and STR of CBD carbamates from pharmacokinetic and pharmacodynamic characteristics.” in line 77.

Line 81, it should be “values were”, also, it should be stated whether those values have been obtained as a part of the current study or they are simply cited

Response: We sincerely thank the reviewer for careful reading. As suggested by the reviewer, we have corrected this mistake in line 82. And those values were obtained as a part of the current study, this was stated in line 84.

Line 86, did you really synthesize the CBD (L0)? Wasn’t it more convenient to purchase this reagent?

Response: Thank you very much for your helpful suggestion, we apologize for this confused expression, and we have corrected the expression in 87-88. The CBD (L0) is natural product, was purchased from Push Bio-technology (Chengdu, China). We have modified the statements as “CBD was purchased from Push Bio-technology (Chengdu, China). The CBD carbamates (L1‒L4), internal standard [IS, L5, CBD 4-bromobenzyl(methyl)carbamate] were synthesized, and the CBD L0‒L5 were provided by School of Pharmacy, Anhui Medical University (Hefei, China)”.

Lines 157-180, it should be clearly stated why L1 and L3 haven’t been compared.

Response: Thanks for your valuable advice. We have clarified this point in line 165-167.

Figure 2C, why there are no error bars for L4? The same applies to Figure 4A.

Response: Thanks for your helpful advice. We have checked the figure, figure 2C for L4 has the error bars. Figure 4A had add the error bars in line 308.

I really appreciate the use of the other (blue) color for the references. However, I’m afraid this will be changed to black at the production stage…
